# Visually Asymptomatic Leaf Loss in *Xylella fastidiosa*-Infected Blueberry Plants

**DOI:** 10.3390/pathogens13100904

**Published:** 2024-10-15

**Authors:** Paul M. Severns, Jonathan E. Oliver

**Affiliations:** 1Department of Plant Pathology, University of Georgia, Athens, GA 30602, USA; 2Center for the Ecology of Infectious Diseases, University of Georgia, Athens, GA 30602, USA; 3Department of Plant Pathology, University of Georgia, Tifton, GA 31793, USA; jonathanoliver@uga.edu

**Keywords:** bacterial leaf scorch, early disease detection, plant disease, disease management, *Vaccinium*, thermal imagery

## Abstract

*Xylella fastidiosa* (Xf), a gram-negative bacterium, is a notorious, world-wide plant pathogen with an extended latent period that presents a challenge for early disease detection and control interventions. We used thermal imaging of tissue-cultured, experimentally Xf-infected blueberry plants to identify visually pre-symptomatic leaves and compared the minimum force required to dislodge symptomatic leaves from infected plants to leaves on uninfected (control) blueberry plants. For two different blueberry cultivars and one pathogenic isolate of *X. fastidiosa*, we found no statistical difference between the mean downward force for leaf dislodgement, regardless of symptom category, on Xf-infected blueberry plants. That force was about 50% to 30% of the mean force to remove leaves from uninfected blueberry plants depending on the cultivar. These results indicate that visually pre-symptomatic leaves may be just as readily lost under field conditions as visually symptomatic leaves, both of which are important for early disease detection. Second, some thermally symptomatic and visually symptomatic leaves appeared to self-prune (abscise) and this may be an unrecognized early symptom of Xf-caused disease in blueberries. Last, it is possible that the self-pruning of visually asymptomatic leaves may occur in other agriculturally and culturally important plants infected by *X. fastidiosa,* but this remains an unrecognized early disease symptom.

## 1. Introduction

The early detection of plant disease outbreaks and rapid treatment are important yet elusive goals for effective plant disease management interventions [1,2,3,4]. Treatment delays during the infectious stage of a plant disease outbreak require disproportionately higher increases in total treatment area to offset even incremental temporal delays [5] which are economically more costly to producers than smaller (potentially one-time only) treatments. Any method that can reliably facilitate early disease detection could be critically important for abating a local disease outbreak or an even more widespread epidemic. For some plant diseases, foliar symptoms can be observed in electromagnetic spectrum bandwidths outside of visible light [6,7] and possibly through volatile compound composition [8]. These non-traditional forms of disease diagnosis often coincide with the simultaneous display of traditional visible foliar symptoms. The early detection of a plant disease outbreak, although clearly important, is particularly challenging for slow-developing plant diseases and those with long latent periods (the time between infection and the expression of detectable disease symptoms) because the symptoms are often subtle and progress incrementally over relatively long periods of time.

*Xylella fastidiosa* (*Xf*) is a gram-negative, insect-vectored bacterium that can cause disease on important perennial agricultural crops such as olives, grapes, coffee, peaches, plums, citrus, and pecans, and infect hundreds of ornamental and native plants, causing disease in some species and not in others [9,10]. Different *Xf* subspecies can be associated with specific host plants and new *Xf*-caused diseases seem to abruptly emerge in warmer regions of the world, such as the case of olive quick decline syndrome in Italy [9,10]. Bacterial leaf scorch of blueberry (BLS) caused by *Xf* subsp. *fastidiosa* and subsp. *multiplex* [11], is a relatively new *Xf*-caused disease in the southeastern US (discovered in 2006) and it leads to significant yield losses and eventually plant death [12,13]. Growers have been observed to remove entire blueberry fields as BLS can intensify to levels where it is more economically sound to remove all blueberry plants and replant with tissue-cultured blueberry plants. Despite initially being recorded in Georgia, Florida, and Alabama, USA, BLS has since been identified in North Carolina [14], South Carolina [15], and Louisiana [13], suggesting that it may be spreading throughout southeastern US blueberry production regions. Because the insect vectors for BLS on blueberry plants have not been confirmed, and antibiotics are completely ineffective treatments [10], the only realistic disease control strategy is the removal of infected plants [16], which requires disease diagnosis based on visually detectable symptoms and verification with molecular methods. However, BLS appears to have a latent period of at least one growing season and potentially up to two or more growing seasons before the visible diagnostic foliar symptoms are presented (Oliver unpublished data). During such an extended latent period, infected but visually asymptomatic plants may be important reservoirs for insect vector *Xf* acquisition and transmission to uninfected plants. Furthermore, infected but asymptomatic transplants and cuttings could lead to new regional outbreaks if latently infected vegetative propagules are used.

Through controlled greenhouse experiments, we found that BLS was detectable using thermal imaging from weeks to months before the expression of traditional diagnostic foliar symptoms [17]. Leaves that were substantially warmer than adjacent leaves, but appeared green and healthy, eventually expressed visual symptoms of BLS suggesting that thermal imaging may be useful for the detection of BLS well before the diagnostic visible foliar symptoms are presented. However, in multiple greenhouse and field container studies over the last several years, including those focusing on thermal imaging [17], we routinely noticed that leaves on experimentally *Xf*-infected blueberry plants would drop green leaves during growing season under optimal growing conditions. Some leaves on *Xf*-infected plants were so sensitive to dislodgement that a slight physical force would cause them to drop from the stems. In some instances, simply rotating the base of a potted plant on a greenhouse bench or transferring a plant to another position was sufficient to induce leaf drop. Some of the dropped leaves displayed foliar BLS symptoms but others appeared to be healthy, green leaves that were visually asymptomatic. The eventual loss of symptomatic leaves was expected as it is a well-known BLS symptom, but the loss of apparently healthy leaves was unexpected. Furthermore, the numbers of asymptomatic leaves dropped, often >10 leaves over a 2–3-day period, were not observed on uninfected control plants. These observations suggested that *Xf* infection may play an important, yet undefined role in the leaf loss of visually asymptomatic as well as symptomatic leaves that could interfere with early disease detection and timely management interventions. 

We conducted a controlled greenhouse experiment to understand whether blueberry plants experimentally infected with *Xf* were more prone to leaf loss than uninfected plants by measuring and comparing the force required to dislodge leaves on plants of two different southern highbush blueberry cultivars (*Vaccinium corymbosum* interspecific hybrids). 

## 2. Materials and Methods

In January 2023, we experimentally inoculated a total of 16 plants (8 plants of southern highbush cultivar ‘Rebel’ and 8 plants of southern highbush cultivar ‘Emerald’) with *Xfm* AR3 [16], as previously described [17]. Eight plants of each cultivar were retained as uninoculated (uninfected) controls. For tissue-cultured experimental blueberry plants, *Xf* inoculum production and experimental inoculations were conducted with the same methods as previously described [17]. 

Plants were monitored weekly for visible BLS symptoms, including subtle leaf yellowing and spotty anthocyanin occurrence (red spots), as well as the later diagnostic symptoms of marginal leaf scorching, yellow and red leaf interiors, and leaf curling (Figure 1). Thermal symptoms were monitored by examining the live images captured with a Fortic 228 (Santa Clara, CA, USA) hand-held thermal camera after a period in an area of exposure to an infrared heat lamp (for details see the methods described in [17]). When we observed either the early visible symptoms of BLS or thermal symptoms (leaf surface temperatures > 2 °C greater than adjacent leaves when exposed for 5 min to a heat lamp) (Figure 1) we monitored disease progression on the plants and conducted the leaf dislodgement experiment ~8 weeks after these initial BLS symptoms (thermal or visual) were expressed. After 8 weeks, blueberry stems adjacent to the inoculated stem were also diseased and there were a suitable range of fully expanded leaves (~20 to 30 leaves/stem) available for experimental removal. Some leaves expressed visibly diagnostic BLS traits and other leaves appeared. 

Cultivar Rebel is highly susceptible to BLS while cultivar Emerald is considered to be tolerant of BLS and also displays more subtle disease symptoms with a slower rate of disease progression [17]. Once we observed that the visible symptoms of BLS began to expand along the inoculated stem and spread to adjacent stems, we conducted an experiment to determine how much downward force (in grams) was required to dislodge randomly selected, fully expanded leaves from uninfected control plants, and two types of leaves on *Xf*-infected plants. On the infected plants, we measured the downward force on visually symptomatic leaves (hereafter ‘symptomatic leaves’) and visually asymptomatic but thermally symptomatic (hereafter ‘asymptomatic warm leaves’). For *Xf*-infected plants, we first located fully expanded leaves with visible BLS symptoms (marginal leaf scorch, interior leaf discoloration of red and yellow areas) [11] and used the thermal camera to confirm that these leaves were also warm (>2 °C warmer than asymptomatic leaves on the same stem or adjacent stems) [17]. Leaves selected for experimental removal were given a small identifying mark with a metallic gel pen and the thermal camera was then used to identify the closest fully expanded leaf on the same stem that appeared to be visually asymptomatic but had the warmth profile associated with the eventual presentation of visible BLS symptoms (Figure 1). These leaves were also given a unique mark with the metallic gel ink pen. For the control (uninfected) plants, we identified a group of leaves that were comparable in developmental stage and position to the *Xf*-infected plants and randomly selected a leaf for experimentation. Leaves on the uninfected plants displayed neither the visible nor thermal symptoms of BLS, so there was no way to compare asymptomatic warm leaves on the uninfected plants as that leaf category did not exist. This method favored *Xf*-infected stems with leaves that were mixtures of visually diagnostic and visually asymptomatic leaves.

To measure the downward force (in grams) required to dislodge the different categories of blueberry leaves from the experimental plants, we used a hand-held, 1000-g push–pull force gauge (Jonard Tools, Elmsford, NY, USA) and attached the hooked end of the gauge on the top of the petiole where it joins the stem. We gently pulled down on the gauge section of the meter until the leaf was removed and recorded the weight (in grams) when the leaf was dislodged. These measurements were analyzed for statistical differences between the control, symptomatic, and asymptomatic warm leaves with a series of *t*-tests assuming unequal variances. Because we had an unbalanced design and were unable to appropriately block the leaves into fully replicated treatment groups, we elected a conservative and straightforward approach for the statistical analysis. An alternative analytic approach would necessitate a significantly more complicated model that would be overfit and unnecessarily complex for the present study.

## 3. Results

For cultivar Emerald, the mean downward force required to dislodge leaves on *Xf*-infected plants did not statistically differ between visually asymptomatic warm leaves and symptomatic leaves (Figure 2). The mean force of leaf dislodgement for leaves on *Xf*-infected plants was approximately one-third of the mean downward force required to dislodge leaves on uninfected (control) plants (Figure 2). However, for the *Xf*-infected plants, the minimal force necessary for dislodgement was <100 g of downward weight (see standard deviation bars in Figure 2).

For cultivar Rebel, the mean downward force to dislodge leaves on *Xf*-infected plants did not differ between symptomatic and asymptomatic warm leaves (Figure 3). The mean force to dislodge leaves on *Xf*-infected plants was about half the force to dislodge leaves from uninfected (control) plants (Figure 3). 

## 4. Discussion

Although we could not perform a full factorial combination of thermal and visual BLS symptoms on *Xf*-infected and uninfected plants, leaves on uninfected plants were significantly more resilient to dislodgement when compared with leaves on *Xf*-infected plants of cultivars Emerald and Rebel. These results are consistent with observations over multiple years of experimentation with *Xf*-infected blueberry plants in greenhouse and experimental outdoor environments. Experimental plants in the present study and also in past studies were provided ample water and fertilizer and were buffered from extreme environmental conditions that could trigger self-pruning in an otherwise healthy plant. Not only was it easier, on average, to dislodge leaves from *Xf*-infected plants, but those differences in force were also meaningful, approximately one-half to one-third the downward force needed to dislodge leaves on uninfected plants. For both cultivars evaluated, the symptomatic and asymptomatic warm leaves did not differ from each other in their mean dislodgement force (Figure 2 and Figure 3). It is highly likely that the true mean value of downward force for the dislodgement of *Xf*-infected leaves was much lower than the values we reported. Multiple leaves on both cultivars of *Xf*-infected plants, both symptomatic and asymptomatic warm, were dislodged from the stem when the hook from the push–pull force meter initially contacted the petiole before the tension meter could be used. Since we could not obtain a measurement for these leaves, we selected the next closest leaf on the same stem in the same leaf category to obtain a downward force measurement. Our past observations and the results presented in this study suggest that asymptomatic leaf loss on *Xf*-infected blueberry plants may be facilitated by minimal external forces (e.g., typical winds, rain, minor physical contact) that would be unlikely to result in leaf loss for a healthy uninfected plant. 

Bare stems (‘yellow twigs’) are a key diagnostic trait for blueberries with BLS [12]. But this bare, yellow stem symptom often shortly precedes either partial or whole plant death [12]. Unpublished field BLS disease progression studies (Oliver) suggest that partial or full plant death takes two to four growing seasons after the expression of the earliest and most subtle of visible foliar symptoms. These early, visible BLS symptoms would not be diagnostic in the field to most growers and would require molecular verification by trained plant pathologists. If *Xf*-infected plants self-prune or more readily drop leaves upon relatively minor disturbances, it is possible that the leaves displaying conspicuous thermal symptoms in the earlier stages of disease progression may also be lost, making early BLS detection more difficult with the reduction in leaves displaying thermal symptoms. We do not yet know the proportion of leaves that are more susceptible to dislodgement and self-pruning on *Xf*-infected blueberries or how pronounced this phenomenon is under field conditions. However, management practices (fruit harvesting, spraying operations, plant maintenance) and field conditions could generate more opportunities for leaf loss compared with the relatively benign greenhouse conditions in the present study. 

It is possible that different *Xf*-infected southern highbush blueberry cultivars and the southeastern US-native *Vaccinium virgatum* (rabbiteye blueberry) may differ in susceptibility to self-pruning and leaf dislodgement. Southern highbush blueberry (*Vaccinium corymbosum* interpecific hybrids) cultivars Emerald and Rebel plants appeared to potentially differ in terms of downward leaf-dislodgement force. The leaves of cultivar Emerald are notably larger, thicker, and have a more substantial waxy cuticle than cultivar Rebel, suggesting a possible role of leaf morphology in the early stages of BLS leaf loss. Anecdotally, rabbiteye blueberry (*V. virgatum*) plants also appeared to drop leaves in response to experimental *Xf* infection, but the diseased leaves rolled up and appeared to self-prune within two days. It is possible that *Xf* may occur in naturally occurring populations of *V. virgatum*, but diseased plants are not detected due to early self-pruning. BLS has not been evaluated in naturally occurring *Vaccinium* populations but there is a surprisingly high amount of genetic diversity within the *Xf* population isolated from single-site southern highbush blueberry fields in Georgia, USA [11,16]. It is possible that naturally occurring *V. virgatum* populations harbor diverse populations of *X. fastidiosa* that spillover into production fields of blueberry cultivars, but this pathogen reservoir is not evaluated because leaves displaying foliar BLS symptoms are quickly self-pruned.

Last, defoliation is a well-known BLS symptom [12] and is also common for *Xf*-caused diseases in other important agricultural crops such as olives [18], pecans [19], grapes [20], and citrus [21]. However, we provide evidence that visually asymptomatic leaves appear to be equally prone to self-pruning when compared with visually symptomatic blueberry leaves. The simultaneous loss of visually asymptomatic and symptomatic leaves is also known for coffee leaf scorch (causal agent *X. fastidiosa* subsp. *pauca*), ultimately leaving the bare stems as an important diagnostic trait [22]. Although the predisposition for the self-pruning (abscission) of visually asymptomatic leaves in *Xf*-infected plants may be limited to just *Vaccinium* and coffee, it may also be possible that *Xf* infection in other important crop plants generates similar patterns of asymptomatic leaf loss early in disease progression. 

## Figures and Tables

**Figure 1 pathogens-13-00904-f001:**
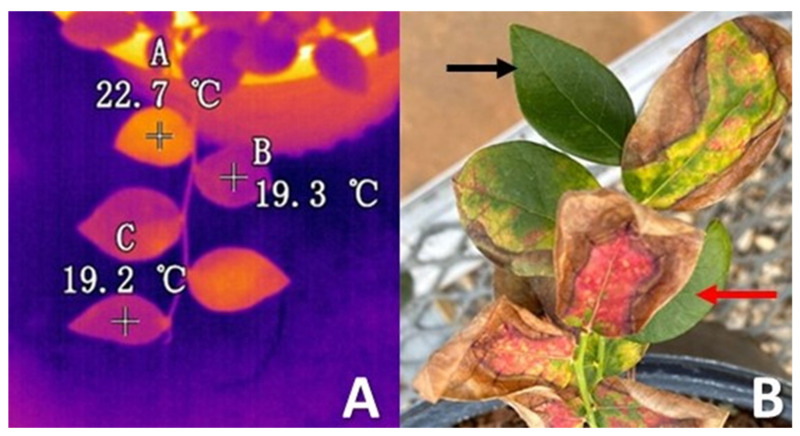
(**A**). Example of blueberry leaves displaying thermal symptoms typical of BLS leaves > 2 °C. (**B**). Leaves with thermal BLS symptoms may be asymptomatic (black arrow) or run a range of symptoms (an example diagnostic BLS leaf is indicated by the red arrow). These figures represent illustrative examples of the different leaf categories and not the actual stems used for testing.

**Figure 2 pathogens-13-00904-f002:**
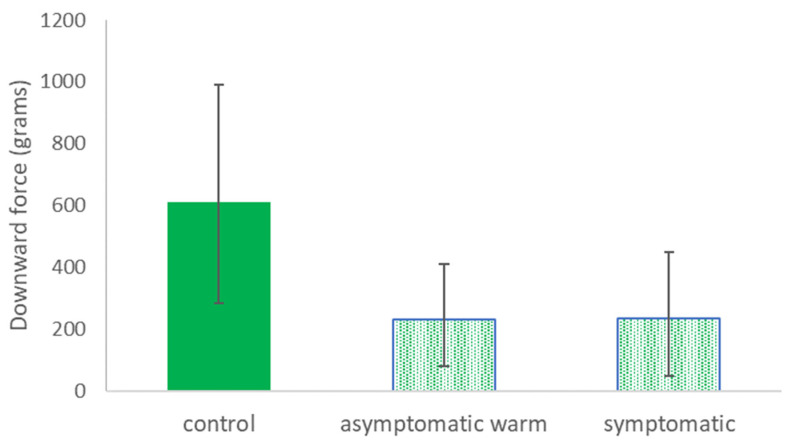
Mean (±st. dev.) downward force (grams) to dislodge leaves from uninfected (control) cultivar Emerald plants, *Xf*-infected plants with leaves that were visually asymptomatic but warm in thermal images (‘asymptomatic warm’), and *Xf*-infected plants with leaves that were visually diagnostic for BLS and also warm (‘symptomatic’). *T*-test results: asymptomatic vs. asymptomatic warm, *n* = 16 plants, t = 0.42, *p* = 0.68; control vs. asymptomatic warm, *n* = 16 plants, t = 8.5, *p* < 0.000001; control vs. symptomatic, *n* = 16 plants, t = 7.7, *p* < 0.000001. Bars with the same fill pattern did not statistically differ from each other according to the *t*-test results reported above.

**Figure 3 pathogens-13-00904-f003:**
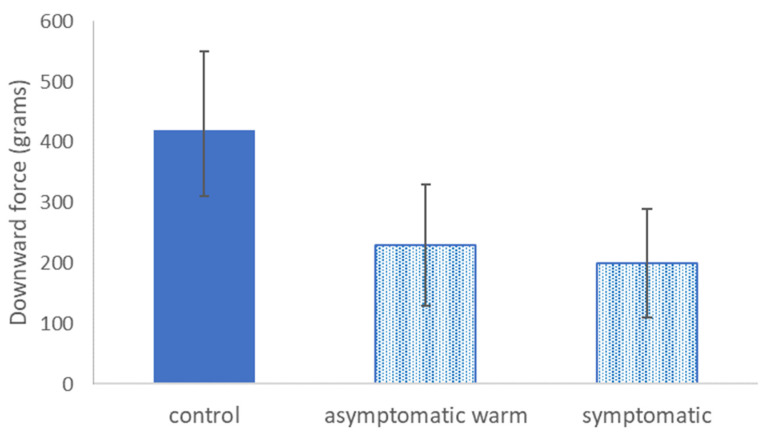
Mean (±st. dev.) downward force (grams) to dislodge leaves from uninfected (control) cultivar Emerald plants, *Xf*-infected plants with leaves that were visually asymptomatic but warm in thermal images (‘asymptomatic warm’), and *Xf*-infected plants with leaves that were visually diagnostic for BLS and also warm (‘symptomatic’). *T*-test results: asymptomatic vs. asymptomatic warm, *n* = 16 plants, t = 0.42, *p* = 0.68; control vs. asymptomatic warm, *n* = 16 plants, t = 8.5, *p* < 0.000001; control vs. symptomatic, *n* = 16 plants, t = 7.7, *p* < 0.000001. Bars with the same fill pattern did not statistically differ from each other according to the *t*-test results reported above.

## Data Availability

Data will be made available upon request to the corresponding author.

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
