# Peer review of "Visually Asymptomatic Leaf Loss in Xylella fastidiosa-Infected Blueberry Plants"

_pathogens, 2024, doi:10.3390/pathogens13100904_

Round 1
Reviewer 1 Report
Comments and Suggestions for Authors
The association of self-pruning to Xf-infection is an original result of the study and could be proposed as a further early symptom of BLS; however, considering that self-pruning also coincide with the appearance of thermal variation and, for some leaves, of symptoms, I doubt that the results of this study can contribute to improving the early detection of the infection.
Minor comment
lane 100: may be the word "adject" is wrong, and the correct one is "adjacent"
Author Response
Reviewer #1
The association of self-pruning to Xf-infection is an original result of the study and could be proposed as a further early symptom of BLS; however, considering that self-pruning also coincide with the appearance of thermal variation and, for some leaves, of symptoms, I doubt that the results of this study can contribute to improving the early detection of the infection.
Yes, we agree the early leaf drop may actually sabotage early disease detection based on thermal symptoms and even detection of BLS in naturally occurring, native populations of Vaccinium spp.. We did attempt to address this conflict in the Discussion section; lines 201 to 232. We have plans to conduct field experiments in the summer of 2025 to understand this issue in more depth.
Minor comment
lane 100: may be the word "adject" is wrong, and the correct one is "adjacent"
We have corrected the typo to adjacent as suggested, thank you.
Reviewer 2 Report
Comments and Suggestions for Authors
The manuscript describes a properly designed experiment and, although it is not accompanied by other experiments in the field and with other varieties and isolates of the pathogen, the results are sufficiently relevant to be published. On the other hand, the methodology followed is very innovative and the monitoring of the data reveals a deep knowledge of the disease by the authors.
Author Response
Reviewer #2
The manuscript describes a properly designed experiment and, although it is not accompanied by other experiments in the field and with other varieties and isolates of the pathogen, the results are sufficiently relevant to be published. On the other hand, the methodology followed is very innovative and the monitoring of the data reveals a deep knowledge of the disease by the authors.
Thank you. We do recognize the need to conduct additional field experiments with thermal imaging and disease status. These experiments are planned for the summer of 2025.
Reviewer 3 Report
Comments and Suggestions for Authors
The authors of manuscript entitled “Visually asymptomatic leaf loss in Xylella fastidiosa infected blueberry plants” used thermal imaging of tissue cultured, experimentally Xf-infected blueberry plants to identify visually pre-symptomatic leaves and compared the minimum force required to dislodge symptomatic leaves from infected plants to leaves on uninfected (control) blueberry plants. Though study could be considered if has some important findings. Moreover, this manuscript is written in not good language; novel study has not been summarized and data presented in the manuscript is not worthy. Further, I have several concern which need to be addressed.
#In abstract and introduction, the paragraph jumps directly into the research without setting up sufficient background. It assumes that the reader is familiar with the significance of Xylella fastidiosa in blueberries and does not provide enough context on the broader implications of the study. It would benefit from a brief introduction that establishes why this research is important.
#The focus of the research is not clear, but the transition between thermal imaging and leaf dislodgement could be smoother. The connection between the two methods (thermal imaging and leaf dislodgement) and how they contribute to understanding Xf's effects is not fully explained.
# The detail materials and methods need to be mentioned, but I would say regrettably that this section is written very poorly and important information is missing.
#In result section, the paragraph presents result but does not fully explain their implications. For instance, the significance of leaves being more easily dislodged from infected plants and how this ties back to disease detection and plant health isn't thoroughly discussed.
# Discussion part is also written in unreadable language. The concluding statements about self-pruning as a potential early symptom and its relevance to other plants are speculative but interesting. However, this idea could be expanded with more detail or supported by additional evidence.
#The mention of "no statistical difference" is good, but the specifics of the statistical tests used, p-values, and sample sizes would enhance the credibility of the findings. Without these details, it's challenging to evaluate the robustness of the conclusions.
#The study is limited to two blueberry cultivars and one isolate of X. fastidiosa. While this is acknowledged, discussing the potential limitations and the need for broader studies across more cultivars and isolates would strengthen the discussion.
#Some sentences are complex and could be broken down for better readability. The passage predominantly uses passive voice, which is common in scientific writing, but occasionally switching to active voice could make the text more engaging.
Comments on the Quality of English LanguageModerate editing of English language required
Author Response
Reviewer #3
The authors of manuscript entitled “Visually asymptomatic leaf loss in Xylella fastidiosa infected blueberry plants” used thermal imaging of tissue cultured, experimentally Xf-infected blueberry plants to identify visually pre-symptomatic leaves and compared the minimum force required to dislodge symptomatic leaves from infected plants to leaves on uninfected (control) blueberry plants. Though study could be considered if has some important findings. Moreover, this manuscript is written in not good language; novel study has not been summarized and data presented in the manuscript is not worthy. Further, I have several concern which need to be addressed.
#In abstract and introduction, the paragraph jumps directly into the research without setting up sufficient background. It assumes that the reader is familiar with the significance of Xylella fastidiosa in blueberries and does not provide enough context on the broader implications of the study. It would benefit from a brief introduction that establishes why this research is important.
Thank you, we have added information about the background of BLS to the Introduction section. We did not feel that the manuscript was worthy of a long paper and tried to keep it shorter rather than longer, but some additional background on the disease would be helpful to the reader which is why we refer the reader to multiple citations in lines 47-61 in the original manuscript. We added the year of discovery to the information in these lines of background, but we are somewhat confused as to what additional information the reviewer would like to read.
#The focus of the research is not clear, but the transition between thermal imaging and leaf dislodgement could be smoother. The connection between the two methods (thermal imaging and leaf dislodgement) and how they contribute to understanding Xf's effects is not fully explained.
We reference a previously published study that explains the link between thermal images and early detection of X. fastidiosa infection in leaves (line 65, citation 17). We did not feel that rehashing this already published and open access study was worthwhile for this manuscript. We are also confused how the final sentence of the Introduction section is unclear concerning our research focus. The sentence reads, “We conducted a controlled greenhouse experiment to understand whether plants infected with Xf were more prone to leaf loss than uninfected blueberry plants by measuring and comparing the force required to dislodge leaves (symptomatic and asymptomatic) on experimentally infected and uninfected plants of two different southern highbush blueberry (Vaccinium corymbosum interspecific hybrids) cultivars.” (Lines 80-84).
However, we substantially reworded the last paragraph of the Introduction to be more straightforward about the history and why early leaf drop is important to understand, especially if visually asymptomatic leaves are easily lost from plants infected with Xf. We also separated the study objectives as its own paragraph. The last two paragraphs of the Introduction now reads:
“Through controlled greenhouse experiments, we found that BLS was detectable using thermal imaging from weeks to months before the expression of traditional diagnostic foliar symptoms [17]. Leaves that were substantially warmer than adjacent leaves, but appeared green and healthy, eventually expressed visual symptoms of BLS suggesting that thermal imaging may be useful for the detection of BLS well before the diagnostic visible foliar symptoms are presented. However, in multiple greenhouse and field container studies over the last several years, including those focusing on thermal imaging [17], we routinely noticed that leaves on experimentally Xf infected blueberry plants would drop green leaves during growing season under optimal growing conditions. Some leaves on Xf-infected plants were so sensitive to dislodgement that a slight physical force would cause them to drop from the stems. In some instances, simply rotating the base of potted plant on a greenhouse bench or transferring a plant to another position was sufficient to induce leaf drop. Some of the dropped leaves displayed foliar BLS symptoms but others appeared to be healthy, green leaves that were visually asymptomatic. The eventual loss of symptomatic leaves was expected as it is a well-known BLS symptom, but the loss of apparently healthy leaves was unexpected. Furthermore, the numbers of asymptomatic leaves dropped, often > 10 leaves over a 2 - 3 day period, was not observed on uninfected control plants. These observations suggested that Xf infection may play an important, yet undefined role in leaf loss of visually asymptomatic as well as symptomatic leaves that could interfere with early disease detection and timely management interventions.
We conducted a controlled greenhouse experiment to understand whether blueberry plants experimentally infected with Xf were more prone to leaf loss than uninfected plants by measuring and comparing the force required to dislodge leaves on plants of two different southern highbush blueberry cultivars (Vaccinium corymbosum interspecific hybrids).”
# The detail materials and methods need to be mentioned, but I would say regrettably that this section is written very poorly and important information is missing.
We reference a previously published study multiple times in the methods section (citation 17) which details all of the methods used in this study with the exception of the measurement of force, which we be believed was explained in enough detail to be replicated. We additionally explained our rationale for the statistical tests (lines 141-147). Our brevity in this section was not due to lack of experimental rigor, but rather the methods have been described in considerable detail in a previous publication. We wanted to avoid any accusations or the appearance of self-plagiarism so we opted to reference the original study.
#In result section, the paragraph presents result but does not fully explain their implications. For instance, the significance of leaves being more easily dislodged from infected plants and how this ties back to disease detection and plant health isn't thoroughly discussed.
Yes, we are following the traditional manuscript structure in life sciences to present but not interpret results in the Results section, but interpret the implications in the Discussion section.
# Discussion part is also written in unreadable language. The concluding statements about self-pruning as a potential early symptom and its relevance to other plants are speculative but interesting. However, this idea could be expanded with more detail or supported by additional evidence.
We reworded most of the Discussion section to be more straightforward and concise with respect to the interpretation of the results and framing their relevance. Thank you for bringing this issue to our attention.
#The mention of "no statistical difference" is good, but the specifics of the statistical tests used, p-values, and sample sizes would enhance the credibility of the findings. Without these details, it's challenging to evaluate the robustness of the conclusions.
The details of the statistical test results were reported in the captions of Figure 2 and Figure 3. We can also report the same test results in the text but feel that the best method is to keep the statistical test results with the figures as all information in contained within the figure and the caption.
#The study is limited to two blueberry cultivars and one isolate of X. fastidiosa. While this is acknowledged, discussing the potential limitations and the need for broader studies across more cultivars and isolates would strengthen the discussion.
Thank you for this suggestion. The second author (Oliver) is presently finishing a manuscript focusing on disease expression in other blueberry cultivars.
#Some sentences are complex and could be broken down for better readability. The passage predominantly uses passive voice, which is common in scientific writing, but occasionally switching to active voice could make the text more engaging.
Thank you for this suggestion. We have gone through the entire manuscript searching for instances where this occurred and have revised the passages to be more concise and straightforward. This helpful suggestion resulted in a rewording of about half of the manuscript. We did not change any of our intent or conclusions, but rather were more straightforward and concise with the writing throughout the manuscript. Because these revisions were extensive, rather than list them all, we highlighted the changes in a marked up version of the revision.
Reviewer 4 Report
Comments and Suggestions for Authors
The manuscript reports early detection of presence of Xylella fastidiosa by very empiric methods. The results and quite expectable however the idea to write a paper about what growers probably know from years by their experience is valuable as teaching to new generation plant pathologist only working in laboratory. An annotated version is provided to eliminate the personalization that must not be used in scientific papers and to provide a bit more scientific information if this is feasible form authors.

In some places English improvement will help to clarify sentences as marked in the annotated version.
Author Response
Reviewer #4
The manuscript reports early detection of presence of Xylella fastidiosa by very empiric methods. The results and quite expectable however the idea to write a paper about what growers probably know from years by their experience is valuable as teaching to new generation plant pathologist only working in laboratory. An annotated version is provided to eliminate the personalization that must not be used in scientific papers and to provide a bit more scientific information if this is feasible form authors.
Thank you very much for the effort in providing corrections to the manuscript. We have considered each suggestion and revised those passages where the suggestion improved the manuscript.
Round 2
Reviewer 3 Report
Comments and Suggestions for Authors
Though this manuscript I recommended reject in the earlier version as there were several critical issues, it came to me again for re-review. I am happy to see the author's reply and I am convinced now with the revised version and also to see their justification for all points raised in the first review. In view of the above and the efforts made by the authors, I agree with the revised version for publication in Pathogens.